# Qualitative exploration of women's experiences of intramuscular pethidine or remifentanil patient-controlled analgesia for labour pain

Victoria Hall Moran ![ORCID] ,[1] Gillian Thomson,[1] Julie Cook,[1] Hannah Storey,[1] Leanne Beeson,[2] Christine MacArthur,[3] Matthew Wilson[4]

¹College of Health and Wellbeing, University of Central Lancashire, Preston, UK
²Birmingham Clinical Trials Unit, The University of Birmingham, Birmingham, UK
³Institute of Applied Health Research, University of Birmingham, Birmingham, UK
⁴Centre for Urgent and Emergency Care Research, University of Sheffield, Sheffield, UK

**Correspondence to**
Dr Victoria Hall Moran;
VLMoran@uclan.ac.uk

## ABSTRACT

**Objectives** To explore women's experiences of remifentanil or pethidine for labour pain and infant feeding behaviours at 6 weeks post partum.

**Design** Qualitative postnatal sub-study to the randomised controlled trial of remifentanil intravenous patient controlled analgesia (PCA) versus intramuscular pethidine for pain relief in labour (RESPITE). Semistructured telephone interviews were conducted at 6 weeks post partum, and thematic analysis was undertaken.

**Setting** Women recruited to the RESPITE trial from seven UK hospitals.

**Participants** Eighty women consented and 49 (30 remifentanil group and 19 pethidine group) completed the interview.

**Results** Eight themes emerged which encompassed women's antenatal plans for pain management (*Birth Expectations*) through to their future preferences for pain relief (*Reflections for Future Choices*). Many women who used remifentanil felt it provided effective pain relief (*Effectiveness of Pain Relief*), whereas women in the pethidine group expressed more mixed views. Both groups described side effects, with women using pethidine frequently reporting nausea (*Negative Physiological Responses*) and women using remifentanil describing more cognitive effects (*Cognitive Effects*). Some women who used remifentanil reported restricted movements due to technical aspects of drug administration and fear of analgesia running out (*Issues with Drug Administration*). Women described how remifentanil enabled them to maintain their ability to stay focused during the birth (*Enabling a Sense of Control*). There was little difference in reported breastfeeding initiation and continuation between pethidine and remifentanil groups (*Impact on Infant Behaviour and Breastfeeding*).

**Conclusions** Qualitative insights from a follow-up study to a trial which explored experiences of intravenous remifentanil PCA with intramuscular pethidine injection found that remifentanil appeared to provide effective pain relief while allowing women to remain alert and focused during labour, although as with pethidine, some side effects were noted. Overall, there was little difference in reported breastfeeding initiation and duration between the two groups.

**Trial registration number** ISRCTN29654603.

### Strengths and limitations of this study

► To our knowledge, this is the first study to explore in depth the experiences and infant feeding behaviours of women using intravenous remifentanil patient-controlled analgesia (PCA) for pain relief in labour.

► Substantial number of women (49) interviewed from multiple sites across England.

► Fewer women recruited who had used pethidine (19) compared to remifentanil PCA (30).

► A high number of women also received Entonox prior to randomisation and in combination with pethidine (41; 84%), and some progressed to epidural analgesia (11; 22%) which may have influenced their experiences.

## INTRODUCTION

Women's experiences of and response to pain during labour are complex, affected by multiple physiological and psychosocial factors, and consequently many women in labour require some form of pain relief.[1] In line with the rise in the use of technology and intervention in childbirth, pharmacological pain management is widely offered to women in labour on maternity/obstetric units. Traditional pharmacological approaches for pain management include opioid analgesics, epidural analgesia and inhaled analgesia (Entonox): 50:50 oxygen and nitrous oxide). Epidural analgesia has been shown to be the most effective form of pain relief for labour but may give rise to unwanted side effects. Systematic reviews of trials have identified that epidural analgesics increase the duration of second stage labour and the risk of instrumental vaginal delivery.[2]

It has been reported that up to 25% of women in the UK use pethidine or a similar opioid during labour.[3] While pethidine is known to alleviate labour pain for some women, when compared with placebo no clear

differences have been observed in maternal satisfaction with pain relief or the number of women requesting an epidural analgesic.[4] Pethidine is known to cause maternal side effects, including nausea, vomiting and sedation.[1] Opioids readily cross the placenta, and neonatal respiratory depression and hypothermia remain concerns.[1] Intrapartum pethidine administration may have a detrimental effect on neonatal behaviour, reducing infant alertness, suppressing the rooting and sucking reflex, and may shorten breastfeeding duration.[5]

There is increasing interest in the role of remifentanil as an alternative opioid analgesia in labour.[6] Remifentanil is a potent opioid medication with a short half-life, administered by patient-controlled analgesia (PCA), allowing the woman to have the drug as and when it is needed, on demand. A systematic review conducted in 2012, based on a few low-quality small trials comparing remifentanil to pethidine administered by various routes, reported a reduction in progression to epidural analgesia with remifentanil.[7] A recent Cochrane review, again including only small poor-quality trials, compared PCA remifentanil to alternative parenteral methods of pain relief in labour, and found that remifentanil (PCA) provided more effective pain relief, and women were more satisfied with pain relief compared to other opioids administered intravenously, intramuscularly or using PCA.[8] Known side effects of remifentanil include respiratory depression, nausea, pruritus, and decreased heart rate and blood pressure, although there are currently too few studies of remifentanil PCA use in labour to draw definite conclusion with respect to side effects for women and newborns.[8] Route of analgesia administration has been found to positively impact on women's experiences of labour, with women who self-administered analgesia (intranasal fentanyl) reporting increased autonomy and satisfaction compared to women who relied on the midwife to administer analgesia, who were more often focused on the physical effect of the drug.[9] The influence of remifentanil on infant feeding behaviours remains largely unexplored.

In this paper, we report on a qualitative follow-up sub-study to an open-label randomised controlled trial undertaken in 14 UK maternity units. The main aim of the trial (RESPITE) was to compare intravenous remifentanil PCA with intramuscular pethidine injection to determine whether remifentanil PCA reduced progression to epidural analgesia; also to consider whether it resulted in any adverse maternal or neonatal sequelae. The trial found that remifentanil intravenous PCA halved epidural analgesia conversion rate, compared with intramuscular pethidine.[10] The importance of combining qualitative evidence that explores the views and experiences of service users alongside quantitative evidence to inform safety, effectiveness and cost of interventions to inform antenatal and intrapartum guidelines has been recognised by the WHO.[11 12] The focus of this follow-up sub-study was to explore women's birth experiences, perceptions of their pain relief and infant feeding behaviours up to 6 weeks post partum, in women allocated to the pethidine or remifentanil trial arms for labour pain.

## METHODOLOGY

### Design

Full details of the RESPITE trial are reported elsewhere.[10] This sub-study follow-up phase was an adjunct to the trial, and comprised a qualitative exploratory study using semistructured telephone interviews.

Participants gave written informed consent before taking part.

### Recruitment

Fourteen UK obstetric-led maternity units took part in the RESPITE trial, where all women booked for delivery at participating centres were informed about the study at their antenatal visits. Women were eligible for the trial if they met the following inclusion criteria: 16 years or older and beyond 37 weeks' gestation, with a singleton live baby, in cephalic presentation, who were in established labour (defined as regular painful contractions irrespective of cervical dilatation), intending vaginal birth and were not participating in any other drug trial. For the qualitative sub-study, women who participated in the trial were approached by a research midwife/nurse after childbirth, while an inpatient, and provided with a separate information leaflet. Women who were willing to take part in a telephone interview up to approximately 6 weeks post-partum were asked to sign a consent form. Participant contact details for consenting women were securely transferred to the research team, with all identifiable data stored on encrypted university files.

Verbal consent was reconfirmed at the time of interview. Women were recruited over a 16 month period (between June 2015 and September 2016), and maternity units that recruited the most women within a 1-month period received a £100 Love to Shop voucher.

### Interviews

Semistructured audio-recorded telephone interviews were conducted between 5 and 8 weeks post partum by two researchers (JC, HS). Telephone interviews provided participants with the opportunity to disclose intimate and sensitive experiences without feeling uncomfortable. The interviews lasted between 20–30 min and explored women's prebirth expectations for pain relief, their experiences and perceived impact of pain-relief on maternal or infant behaviours, and infant feeding. Questions such as 'How did the pain relief affect your labour? (eg, emotionally, physically)', 'How did you feel after the birth?' and 'How was your baby following the birth?' were posed. All women who took part in an interview received a £10 Love to Shop voucher to thank them for their participation.

### Analysis

All interviews were transcribed verbatim, anonymised and uploaded into NVIVO qualitative software.[13] Data

collection and analysis was undertaken concurrently, and Braun and Clark's thematic approach[14] was undertaken to identify patterned meaning across the dataset. Two authors (JC, HS) undertook this work (under the guidance of VHM and GT) through an iterative process of reading, identifying key codes, grouping codes into sub-themes and finally creating themes that were reflective of all views expressed. We paid particular attention to how women's experiences of pethidine and remifentanil, within the context of the trial, compared or differed. Analytical decisions were discussed and shared with all authors both periodically and at the end, to agree on final themes.

## Patient and public involvement

Members of the Public and Researchers Involvement in Maternity and Early pregnancy group, a group of maternity service users convened by the University of Birmingham, were involved in reviewing the participant information during study set-up and were represented on the Trial Steering Committee for the RESPITE trial.

## RESULTS

Eighty trial participants from seven hospitals agreed to take part in the RESPITE postnatal sub-study. While all 14 sites participating in the RESPITE trial were eligible for the sub-study, only seven hospitals successfully consented participants who progressed to interview. Four maternity units received £100 in shopping vouchers as reward for recruiting the most women within a 1 month period. Despite numerous contact attempts, 23 women were not interviewed due to women being unavailable during the timeframe of the study or due to language barriers. Eight women asked to be withdrawn from the study prior to interview. Forty-nine participants (30 remifentanil and 19 pethidine) took part in a telephone interview and were included for analysis.

Demographic information of participants is provided in table 1. Eight women received pethidine before they entered the trial (trial eligibility stipulated no systemic opioid pain relief within the previous 4 hours), four of whom were subsequently assigned to the pethidine trial arm, and four to the remifentanil arm. The majority of women (41/49) received gas and air (Entonox) pain relief in addition to the trial drug. Overall, 11 women progressed to epidural analgesia: seven from the pethidine arm (conversion rate of 64%) and four from the remifentanil arm (conversion rate of 36%), somewhat higher than the conversion rates seen in the main trial (41% and 19% for pethidine and remifentanil respectively).[10] Of the 11 women who had an epidural analgesic, five women had a caesarean section; two from the pethidine arm and three from the remifentanil arm. A further three women had a caesarean section without having had epidural analgesia.

In the following section, we report on eight interlinking themes, relating to women's expectations and

**Table 1** Demographic information of study participants

| | Remifentanil (n=30) | Pethidine (n=19) | Total (n=49) |
|---|---|---|---|
| **Age of women (years)** | | | |
| Mean (SD) | 29.3 (5.2) | 25.9 (4.5) | 28.0 (5.2) |
| Range | 19–39 | 17–34 | 17–39 |
| **Ethnicity** | | | |
| White | 23 | 14 | 37 |
| Asian (Indian) | 2 | 1 | 3 |
| Asian (Pakistani) | 2 | 4 | 6 |
| Mixed | 1 | 0 | 1 |
| Other | 2 | 0 | 2 |
| **Marital status** | | | |
| Married/living with partner | 30 | 17 | 47 |
| Single | 0 | 2 | 2 |
| **Parity** | | | |
| Nulliparous | 21 | 13 | 34 |
| Parous | 9 | 6 | 15 |
| **Type of birth** | | | |
| Vaginal | 26 | 15 | 41 |
| Caesarean | 4 | 4 | 8 |
| **Additional pain relief administered** | | | |
| Epidural analgesia | 4 | 7 | 11 |
| Pethidine | 4 | 4 | 8 |
| Entonox | 24 | 17 | 41 |
| **Infant feeding intention** | | | |
| Breast feeding | 21 | 12 | 33 |
| Formula feeding | 6 | 6 | 12 |
| No prebirth plans | 3 | 1 | 4 |
| **Infant feeding initiation within the first hour postbirth** | | | |
| Breast feeding | 14 | 8 | 22 |
| Formula feeding | 16 | 11 | 27 |
| **Infant feeding status at ~6 weeks** | | | |
| Exclusive breast feeding | 7 | 0 | 7 |
| Formula feeding | 16 | 14 | 30 |
| Mixed feeding | 7 | 5 | 12 |

experiences of pain relief, and the impact of analgesia on women's experiences of infant feeding across the cohort; with similarities and differences in the experiences of different forms of pain relief highlighted. As no stark differences were found between the responses of nulliparous and parous women, no differentiation was made in the findings. A selection of participant quotes are included, together with an identifier that reports that participant number, and either an R (remifentanil) or P (pethidine) suffix to indicate their group allocation.

The eight themes identified encompass women's plans for pain management in the antenatal period (*Birth Expectations*), through to how their experiences of using either remifentanil or pethidine may influence their future

preferences for pain relief in labour (*Reflections for Future Choices*). Four themes centred around women's physiological and cognitive responses to the analgesics (*Negative Physiological Response; Cognitive Effects*), and reflections on how effective pethidine and remifentanil were for pain relief (*Effectiveness of Pain Relief*) or pain management (*Enabling a Sense of Control*). Additional themes highlighted common issues with the administration of the analgesics which may have impacted negatively on women's experiences (*Issues with Drug Administration*), and their reflections on whether they felt the analgesics had an influence on their newborns and the establishment of breast feeding (*Impact on Infant Behaviour and Breastfeeding*).

### Birth expectations

There were wide variations in women's approaches to birth planning. These ranged from some women having no clear plans; 'there wasn't any plans, no' (14R) or how women 'just went with the flow' (36R), to having very specific intentions; 'Yes I did a lot of planning, I knew exactly what I wanted' (50R). Often women's approaches to labour were related to their previous childbirth experiences. Prior experiences of unfulfilled birth plans often led women to adopt a more flexible perspective due to the uncertainty of childbirth:

> Last time I was determined I wasn't going to have an epidural and then things didn't quite go according to plan so I had an epidural, so this time I decided to be, you know to go more with an open mind, because things can't be planned as much as you would like. (36R)

The lack of a specific birth plan did not mean that women held no preferences or opinions regarding pain relief. Many women had actively researched pain relief options in a variety of ways, such as through antenatal classes, internet searches, and discussions with friends and family, although only two women had any prior knowledge of remifentanil. Some women considered pethidine to be the best available option:

> Well I'd looked at all the different types of pain relief beforehand and decided it was the one that I thought would be the best, so the pethidine. (15P)

However, others were less inclined to use pethidine due to the potential impact on the baby:

> It was really just the fact that it [pethidine] crossed over [to] the baby, and like I didn't really want him coming out sleepy. (55P)

Almost all the women interviewed were initially opposed to having an epidural analgesic due to reasons such as needle phobia, prior negative experience of epidural analgesia and vicarious reports of risk:

> I'd heard so many stories of people having nerve damage and complications. (16P)

For a few of the women, their reticence reflected cultural expectations of how childbirth should be managed:

> I think it's a culture thing. Like, originally my parents are from [country], they're very against it [epidural]. I don't know if it's a myth or if there's any, or there is any research to back it up but they say it can cause lots of problems in the future and stuff like that. So we've kind of just been brought up around that. I've always kind of been told, 'Oh yeah, you know don't get that'. (33P)

### Negative physiological responses

Women from both trial arms reported what they considered to be side effects and negative impacts of the analgesia on their physiological responses. While pethidine has a well-known sedative effect, for some of the women in our study this was not related to effective pain relief, and rather became problematic when it rendered them incapable of staying awake in labour:

> Well I just remember them giving me an injection and […] it just made me, literally just trying not to fall asleep, that's how it felt. Like my eyes were really heavy and I were just nodding off but I was still in so much pain it just made things a bit worse if I'm honest because it didn't feel like it was numbing the pain at all it just literally made me want to go to sleep and I couldn't. […] … (73P – progressed to epidural analgesia)

Some women across both trial arms experienced drowsiness to the point that it interfered with their autonomy and mobility during labour:

> I was lying down and they told me not to move too much because obviously because it makes you that drowsy and that sleepy that it's best to stay in one position and just to press a button when you feel the pain. (45R)

Many of the women in the trial already had their mobility restricted by monitors, oxytocin infusion and so on, and some of these women found that the delivery of remifentanil through a PCA device requiring continuous intravenous access further restricted their mobility:

> I found it [labour] very painful and I couldn't walk around, I couldn't really be on all fours very easily. All the things that I had imagined, I just couldn't do because I was attached to the machine as well so yeah […] (65R)

Several women in the pethidine arm reported experiencing vomiting or nausea, compared to only one woman in the remifentanil group. However, this woman reflected that it was difficult to disentangle the effects of remifentanil from the side effects of other analgesia:

Perhaps a little bit drowsy because I know I was a bit in and out, and I was quite sick but I don't know if that was gas and air more than the actual drug. (74R)

Furthermore, for some women, their adverse physiological responses to remifentanil were only experienced when the analgesia 'ran out':

The only thing I would say is because I was on it for quite a long time, the drip actually ran out and when the drip ran out, I had quite bad sickness and like body-shaking, my whole body was shaking, also it kind of, I think it got blocked at one point as well so that happened twice where like I was violently throwing up everywhere. (60R)

On some occasions, as reflected in the following quote, symptoms associated with remifentanil led some women to cease using this form of pain relief:

…the first couple of times I did it okay, from the first go I didn't, I said I don't feel right and then after that I couldn't focus enough to even press the button when I needed to… I was just not with it, and then my hubby said, 'Oh can I press it for you?', but the midwife said, 'Oh no it's got to be…', it had to be me that did it. But I just couldn't, so I think in total I only pressed it probably four or five times and then decided to stop because I just didn't, I just didn't like it at all. (36R)

### Effectiveness of pain relief

There were mixed views regarding the effectiveness of pethidine for pain relief. Some women were happy with the pain relief obtained from pethidine; 'it was brilliant. I literally didn't feel a thing until after she was born' (38P). Women were satisfied with its effectiveness, particularly when used in combination with other pain-relief methods, for example, gas and air:

When I just had the gas and air it just felt like, wasn't really working at all, wasn't really doing anything but then obviously the two together just sort of took the edge off a little bit. Obviously I remember still feeling a bit of pain but it's sort of bearable if you like, a bit more bearable. Yeah, definitely felt the difference. (52P)

From a counter-perspective, half of the women who received pethidine reported that their pain relief had been ineffective. These women reported about how they could 'feel everything' and how it had made 'no difference' to the intensity of their contractions:

I didn't find the pethidine any help. It didn't help me at all […] like, she said that it would like ease the contractions off, you know like the pain, but I didn't feel, I could still feel the pain when she, after the pethidine got injected in I could still feel everything […] It might help other women but it didn't help me. (79P)

In contrast, only a few women found remifentanil ineffective as pain relief. However, this minority of accounts were generally related to delivery of the medication or its inability to prevent pain caused by the baby's position in utero:

As the labour progressed, because my son was back-to-back, a lot of the pain was in my back and I found it helped with the contractions but not necessarily the back pain in between the contractions. (58R)

Overall, there was a notable enthusiasm for remifentanil among participants. Some women described experiencing 'no pain' after its administration, whereas others referred to how it made the pain more bearable:

Yes, it did. It definitely did help because before that it was just getting really unbearable. It was just so unbearable, I just couldn't, I felt as if I wanted to die [laughs], I know that's an exaggeration but that's how it feels at that point where you just can't bear that pain but when I took that medication or pain relief, it took that pain away. Obviously as soon as I felt that contraction coming, press the buzzer and it's like more than half of it were gone and it were just, it just slowly eases it all off and it were quite good. It was bearable then. (59R)

### Enabling a sense of control

One of the key benefits of remifentanil highlighted by women was how it provided pain relief *and* maintained their ability to stay alert, awake or focused during the birth:

I still remember being very alert and having to listen to what the midwives were saying, you know push and everything, so I was still very alert and able to do what I needed to do […] (44R)

For some women, being connected to a cannula did not substantially restrict their ability to move from the bed during labour, which, as reflected in the quote below, could have a positive impact on women's agency and self-efficacy:

That's what I wanted to do anyway. I wanted a water birth. I wanted to be active. As soon as I got into labour anyway, before I had that medication, I didn't want to be on the bed, I hated being on the bed, being stuck with, restricted. I didn't like it, I just wanted to be able to move and it started giving me a little bit more confidence as well, you know, I just, not even, obviously I didn't care, I just wanted to get her out. I couldn't have done it without that [remifentanil], I really couldn't. (50R)

Women often needed time and practice to get into the rhythm of delivering remifentanil via the PCA. Once mastered, several women talked about how the PCA enabled them to retain a sense of control by actively managing their own labour:

But it was just so nice that you could be in control of it and it was one of those things that I didn't have to have it all the time. So, for example, with contractions, I'd press the button on the first, you know, with one contraction, and then I wouldn't have it for a couple of times because I'd be in control of my body, if that makes sense. […] And for me, it was just what I wanted, I could feel, I was in control of me, I was in control of my body. I knew what was going on in my surroundings. (1R)

This capacity for self-regulation was supported by the immediate impact of remifentanil on women's physiological responses:

But I noticed as well, when I pressed the buzzer, it worked straightaway, you know, the pain… […] I just noticed like when I pressed the buzzer, you know, when I pressed the buzzer, you know, for pain relief, it was like instant, yes, the pain just like went. (12R)

The ability to maintain focus was strongly linked to the short-acting effect of remifentanil. The fact that each dose of remifentanil wore off quickly was experienced as empowering. One participant found that discontinuing use of the PCA quickly re-established the control she felt she was lacking:

I really feel like when it came to the time of needing to push and things, when you're having to do sort of certain things at certain times, I needed to, I would have struggled if I had still been taking it because I couldn't kind of concentrate on anything, whereas because I'd stopped it and it was pretty much out of my system I was able to focus more. So I don't think it was in my system for very long. (36R)

On occasion, these positive effects were contrasted with women's previous experience of using pethidine:

It was, obviously, because you've got control of it. So, you know, you were able to press it when you needed it. Whereas like, if you have the pethidine and that, once you've been given it you've got no control over how it's going to work. (14R)

While some of the women who had received pethidine made positive comments about its impact on their sense of control during labour, this was often in the context of providing respite to manage their labour pains:

So I had the pain relief for a good few hours which had kind of given me a bit of respite actually because I was getting quite tired and quite, I really needed pain relief so when I got that then it just gave me a little bit of respite…(6R)

## Cognitive effects

Some women, particularly in the remifentanil group, reported cognitive effects consistent with systemic opioids. For some, this was a negative experience, associated with a sense of vulnerability, where they felt unable to focus or process external stimuli and could, as reflected below, negate the potential salutary effects of medication use on labour pains:

I was, I don't really know the technical term, almost knocked out by it, it made me really sort of, I don't know if that's the right word but I was really out of it, I couldn't focus, it was almost […] like it was too strong for me and they reduced the dose of it but […] the midwife was trying to ask me questions and give me instructions as they do, it's ongoing, and I really couldn't […] focus because I was almost sort of not in the room as it were, so for me that sort of, sort of negated any positive effect it would have had on the pain relief side of things. (36R)

Whereas while other women described feeling 'out of it' or 'away with the fairies' after remifentanil administration, this was not always seen as being entirely negative:

I would say it did take away my control and my sort of presence in the process so, you know at one point I could hear my mum and husband talking about me but couldn't voice an answer to them…Now that did mean that I was, you know, in a lovely place, and I wasn't feeling any pain at that point but it meant that I was almost removed from the labour process. (58R)

A number of women in the remifentanil group also reflected on how their confused or altered state of mind was observed by their birth partners:

I mean my husband said that I was talking about the car insurance, I was saying to him, 'Is the car insured?'…I was having a conversation about the car insurance and he was like, this is a weird time to talk about the car insurance. (56R)

I was in, kind of in a completely different zone I think. My mum and husband were there and said that I'd gone from being in incredible pain to just really blissed out. (58R)

For both pain relief medications, women likened the effects of the analgesia to heavy alcohol consumption:

To be honest I felt incredibly drunk or to the same effect. (55P)

It was like being very, very, drunk. (76R)

A further negative impact reported by women with respect to both forms of pain relief related to how the analgesia had impacted on their memories of labour and childbirth:

I must admit within, literally within, I've no idea because I was out of it. […] So then, again I don't know how long I was on it for. […] Or maybe she was crying. I can't remember. I can't remember, sadly. (44R)

## Issues with drug administration

On some occasions, difficulties were experienced due to the potential or actual unavailability of remifentanil. For example, some women described feeling 'stressed out' due to concerns about whether the analgesia (drip bag) would run out:

> The pain relief it was brilliant, really, really good for about 3 hours and then the effects started to wear off. I don't know if the contractions were getting worse but I was clicking it a lot more than I had originally and I was running out of it and I was getting a bit stressed out because I was worried that I was going to run out. (32R)

In some instances, remifentanil was only reinstated after what women felt was a long delay (if at all), leaving them unexpectedly without pain relief:

> …but to leave someone without, there probably was a bit of pain relief in me but that ten minutes was a long ten minutes. Yeah, it's quite traumatising. (63R)

One woman reported that the issue of remifentanil running out had contributed to her using epidural analgesia:

> I ended up going on to have an epidural because my medication had finished out and what happened was my baby's heart rate kept dipping in and out. So they were a bit concerned and, obviously, I was still in pain after that, then I wasn't fully dilated. So I needed something to, to take the edge off, basically, so I did go in for an epidural, yes. (7R)

Several women also reported difficulties in opportunities to receive pain relief when the cannula became detached:

> I'd managed to rip my thingy out, the cannula thing so I didn't have it through the last bit basically because she said, 'Oh I can't get it put back in, you'll just have to go on gas and air for the last bit'. (44R)

For some women, the physiological effects of the remifentanil interfered with their ability to use the PCA to deliver the pain relief for each contraction:

> The other thing was, when I was using the button this is, it goes straight to your head, you know, and I was so woozy, when my next contraction would come, I would forget to press the button. And that contraction would feel absolutely dire because obviously the contraction before it had taken the edge off, and you know the midwife had to keep telling me [name] you need to press the button, and I was like, oh OK, OK. But I was in and out of it if I'm honest, but it was good, it was good. (48R)

Whereas for others, as reflected in the quote below, the stability of the cannula directly affected their mobility:

> When I lifted up my hand it never went in properly, it had come up with an error message. So I had to make sure my hand was down when I pressed the button, which is a bit difficult when you're in contractions… No, I told them it wasn't working but they just told me to keep my hand down. (10R)

## Reflections for future choices

Some of the women who received pethidine reported that they would opt for this form of pain relief in a future birth, even where the analgesia had not been wholly effective:

> It didn't get rid of the pain, but I would have it again…Yes, I wouldn't even think about it, I'd just have it next time…Obviously I'd go for the gas and air again if I could and then if it was like I was told I couldn't have it then I would go for the pethidine. (69P)

In converse, others were adamant that they would not use pethidine again, that is, 'I hope I never have to have it [pethidine] again' (55P), and would where possible choose an alternative such as remifentanil:

> If hell froze over and I got pregnant again, I'd definitely have, I'd request remifentanil from the get go. (25P)

Similar mixed responses were also evident among women who received remifentanil. A number of women referred to how they would 'definitely' use it again:

> I definitely felt I needed it so I would definitely have it again and I would definitely choose the same form of pain relief if it was offered to me again. I felt like it really worked for me. It definitely gave me what I needed and what I wanted from pain relief so I would definitely do the same again. (39R)

And would recommend it to others:

> I've also said to any other friends…if you are offered it [remifentanil], go for it because it's brilliant. (1R)

Whereas other women, notably all of whom had experienced adverse responses, would either request pethidine or make concerted efforts to not use pain relief:

> Gas and air, yeah I'd have gas and air again, unless I needed it epidural no. I don't think any of the trial drugs, or the drug I had before I wouldn't have. So just gas and air…But me, I don't think I'd do it again. It didn't agree with me, and yeah I don't think I'd take it myself. (29R)

## Impact on infant behaviour and breast feeding

The majority of women (33/49) expressed a prior intention to breast feed their babies, while 12 women intended to formula feed and four had made no plans prior to their baby's birth (see table 1). Within the first hour of birth, 47% (14/30) of women in the remifentanil group

and 42% (8/19) of women in the pethidine group breast fed. By the time of interview (approximately 6 weeks postnatal), 47% (14/30) of women in the remifentanil group and 26% (5/19) in the pethidine group were either exclusively breast feeding or mixed feeding their infants. A number of reasons were given for an earlier than expected cessation of breast feeding, including the level of support received, feeding difficulties such as infant tongue-tie and inverted nipples, and health problems for the mother (eg, lupus, heart condition, mental health issues) or the baby (need for antibiotics, cleft palate, inadequate weight gain). None of the participants felt that the pain relief medication they received influenced their decision to cease breast feeding.

Women in both groups reported feeling exhausted immediately following the birth, and how this affected their initial interactions with their baby. However, as the effects of analgesia, and particularly remifentanil, dissipate after a short period of time, these complaints are likely to be attributed to the length and exertions associated with labour, rather than pain relief analgesia per se:

> The first day, obviously, the night that he was born, I didn't [have skin to skin], after he was born I was just asleep. When my mum come that evening, that afternoon, so she was there and my husband was there, and that day I was asleep. (11P)

> Uncomfortable, tired, drowsy like even the nurses had to take over sometimes to feed my son in the night because I was that drowsy still and I couldn't, I just couldn't take any control, it was a case of I was falling asleep with him on me and stuff like that because I was just so tired. (17R)

Other women felt so weak that they needed help from their birth partners to hold their babies:

> So he was born, they were saying, give him milk, they put him on to my breast and I was giving him the milk but my arms didn't have that much strength to hold him, my husband had to help me out. (11P)

One woman who had progressed to epidural analgesia reflected on her increasing frustration at not being able to breast feed her baby:

> I was so exhausted. I couldn't really move because of my epidural and I was getting really tired and a bit emotional and everything so it wasn't, breastfeeding wasn't working basically. (44R)

## DISCUSSION

This study aimed to explore women's experiences of remifentanil and pethidine use for pain relief during labour and infant feeding behaviours up to approximately 6 weeks post partum. The findings revealed that while many women had actively researched pain relief options prior to their birth and had either formulated birth plans

or had decided to adopt a more flexible approach to pain management, most were unfamiliar with the opioid remifentanil until they were informed about the trial. The majority of women who had used remifentanil felt that it had provided effective pain relief. This was not the case with women in the pethidine group, who had more mixed views, with many reporting that pethidine was completely ineffective at relieving their pain. This was consistent with the findings of the RESPITE trial, which showed that women's views of their pain relief effectiveness and average pain scores were significantly better in the remifentanil arm.[10] These insights also support a recent study by Fleet and colleagues,[9] who compared women's use of pethidine and fentanyl (delivered on an intranasal or subcutaneous basis). Fleet and colleagues' study found that women who received fentanyl provided more positive, and less negative responses compared to those who received pethidine.[9] However, it is important to note that while effective pain management has become an essential component of the care plan for expectant mothers, women's perceptions of pain are affected by physiological and/or psychological issues (eg, fear, anxiety),[15] as well as the nature of the woman-provider relationship.[16]

Women using either form of pain relief reported experiencing a number of side effects, although these tended to differ according to the analgesia used. Women using pethidine frequently reported feeling nauseated, consistent with the findings of Fleet and colleagues,[9] whereas this was only reported by one woman who had used remifentanil. Women in both groups reported that the opioid made them feel drowsy, although this seemed to be less pronounced in the remifentanil group. These insights support the RESPITE trial findings[10] and the work of others such as Volikas and Male[17] with regard to the sedative effects of opioid-based forms of pain relief.

A number of negative comments arose relating to the nature of remifentanil administration. Women in the remifentanil group described how the PCA restricted their movements, and for a small number this resulted in pain (as a result of technically flawed intravenous cannulation) and frustration. Others experienced anxiety or difficulties due to the potential or actual unavailability of remifentanil. Women reported feeling anxious about whether the analgesia (drip bag) would run out, and on occasions when it did, they experienced significant delay in its reinstatement. On some occasions, women considered that this may have contributed to their progression to epidural analgesia. Other qualitative researches have also highlighted how opioids administered too late, or which wore off too early, created fear and anxiety.[9 18] It is important to note that these technical difficulties are not inevitable components of PCA and are amenable to resolution.

One of the key advantages of remifentanil highlighted by women was how it provided pain relief and maintained their ability to stay focused during the birth. Several women referred to how PCA enabled them to use pain relief selectively, as required. In this sense, they

saw remifentanil as an opportunity to stay in control and actively manage their own labour (not just the pain), helping them to find reference points for their own understanding of their progression and tolerance to pain. The ability to maintain focus was probably related to the short-acting effect of remifentanil which enhanced their sense of control. The association between the use of pharmacological pain relief and increased sense of control has also been reported by others.[9 12 19 20]

A further notable finding, as reflected by others,[9 21] relates to women experiencing a cognitive impact from the analgesics. Some women reported feeling disconnected from their labour and unable to focus or process external stimuli. Similar findings were reported in a study by Whitburn and colleagues to explore women's experiences of labour pain. They found that an inability to retain focus during labour was associated with self-judgement and negative perspectives of pain.[22] Some women felt that remifentanil impacted on their memory and affected their ability to recall their baby's birth. While this was not found in Fleet and colleagues' study,[9] these differences may be related to additional factors such as levels of fear, and dosage of the medication, or the fact that more women in our study received remifentanil than pethidine.

In relation to infant feeding, the main trial found no difference in breast feeding within an hour of birth in the two trial arms[10]; with a similar pattern noted among those in this sub-study. Forty-seven per cent of women in the remifentanil and 42% of women in the pethidine arms initiated breast feeding within the first hour after birth. At around 6 weeks postnatal, the rates of 'any' breast feeding were 26% for women who used pethidine compared to 46% of women in the remifentanil group. While these variations may be attributable to the level of postnatal support for breast feeding across the different areas, the findings do support those by Fleet and colleagues[23] in that women who used remifentanil had fewer breastfeeding difficulties. These differences may relate to the differential effects of analgesia on infant behaviours which may impact on establishment of sustained breast feeding.[24] However, as there are many biopsychosocial factors that can influence the mother's infant feeding decisions, further research is needed.

To our knowledge, this is the first study to explore the experiences of women using intravenous remifentanil PCA for pain relief in labour in depth, by formal, qualitative methodology. The study comprised a qualitative sub-study to a multicentre randomised controlled trial (RCT) that compared intravenous remifentanil PCA with intramuscular pethidine for pain relief in labour. The strengths of this study are that to date, there is minimal qualitative research to understand women's experiences of opioid use during labour, and the substantial number of interviews conducted in multiple sites across England. The limitations relate to the relatively fewer women recruited who were randomised to pethidine, and it may be that women who held stronger views towards the use

of analgesia (positive or negative) took part in the study. The high attrition rates for the study highlight challenges intrinsic to early recruitment into postpartum studies. For example, women may not have been fully cognisant of what they were agreeing to, and difficulties are encountered in organising data collection when participants are dealing with the demands of a new baby. It is also important to reflect that 41 (84%) of our sample received Entonox prior to randomisation and in combination with pethidine, and 11 (22%) progressed to epidural analgesia, which may have influenced their experiences.

## CONCLUSIONS

Qualitative insights from a follow-up sub-study to an open-label trial that compared intravenous remifentanil PCA with intramuscular pethidine injection highlighted valuable differences. Overall, more women found remifentanil to be an effective pain relief compared to the mixed views of women who received pethidine. All women experienced side effects although these differed between groups, with a number of the difficulties associated with remifentanil related to drug administration rather than impact of the analgesia. While negative experiences of remifentanil included restricted movements due to drug administration, a key advantage of this form of pain relief was how its short-lasting nature enabled women to remain focused during the birth. Overall, while there was little difference in breastfeeding initiation rates, there was an indication that women who received remifentanil were more likely to continue breast feeding. While these variations may be attributable to different numbers of women included from the trial groups and the level of postnatal support for breast feeding across the different geographical areas, it may be that women who used remifentanil had fewer breastfeeding difficulties.

**Acknowledgements** Professor Jane Daniels contributed to the trial design and conduct. We thank all the women who participated in the RESPITE postnatal sub-study and thank Dr Karin Cadwell and Kajsa Brimdyr of the Healthy Children Project for their support in funding this study. Finally, we express our gratitude to our National Health Service colleagues who supported recruitment for the study.

**Contributors** The postnatal sub-study was conceived by VHM and GT and they secured funding to conduct it. VHM and GT performed the literature search and were responsible for the study design, with input from MW, CM and LB. Study conduct and data collection was led by VHM, GT, JC and HS with contributions from MW, CM and LB. Study analysis and was performed by JC and HS with supervision from VHM and GT. All authors were involved in data interpretation. Writing of the paper was led by VHM, GT, JC and HS with input from MW, CM and LB.

**Funding** This work was supported by the Healthy Children Project, Inc. The RESPITE trial was funded by the National Institute for Health (NIHR) Research Clinician Scientist Award (NIHR ref: CS-11-030). CM is part funded by NIHR Collaborations for Leadership in Applied Health Research and Care West Midlands.

**Disclaimer** The views expressed are those of the author(s) and not necessarily those of the NIHR or the Department of Health and Social Care.

**Competing interests** None declared.

**Patient consent for publication** Not required.

**Ethics approval** Ethics/governance approval was obtained from a National Research Ethics Service Committee (NRES Committee East Midlands - Nottingham

2, reference number: 13/EM/0239), the lead authors' University (STEMH 304), and via the participating Trusts' R&D departments.

**Provenance and peer review** Not commissioned; externally peer reviewed.

**Data availability statement** The data (deidentified quotes) are available from the corresponding author upon reasonable request.

**ORCID iD**
Victoria Hall Moran http://orcid.org/0000-0003-3165-4448

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
