## [Reviewer comments · BMJ Open]

ARTICLE DETAILS

TITLE (PROVISIONAL)	A qualitative exploration of women's experiences of intramuscular pethidine or remifentanil patient controlled analgesia for labour pain.
AUTHORS	Moran, Victoria; Thomson, Gillian; Cook, Julie; Storey, Hannah; Beeson, Leanne; MacArthur, Christine; Wilson, Matthew

VERSION 1 - REVIEW

REVIEWER	Dr Laura Whitburn La Trobe University, Victoria, Australia
REVIEW RETURNED	24-Jul-2019

GENERAL COMMENTS	Thank you for giving me the opportunity to review this manuscript. Women's experiences of labour pain and pain interventions is of great importance and the findings can contribute towards the care of women through labour and birth. This manuscript has the beginnings of a valuable paper, but I feel that it still needs some refinement, particularly regarding the presentation of the results and the discussion. Intro: The first paragraph of the intro could better focus the topic of the paper. I think that the first sentence, whilst VERY important, seems a bit out of place as it is not well integrated with the next. You may wish to work on this. The objective may also need to be reconsidered / rewritten. As it currently reads (in the abstract and in introduction), it sounds like you are exploring women's birth experiences of pain relief and infant feeding intentions and behaviours – however of course infant feeding intentions and behaviours are pre- and post-natal experiences respectively. I think that the primary aim was to explore women's birth experiences of the two different forms of pain relief. However, the aim to explore women's infant feeding intentions and behaviours does not quite fit – there is no mention of the relationship between pethidine/remifentanil and breastfeeding at all in the introduction. You may need to make the decision to focus only on the pain experience, or, if you intend to include the data on feeding, this should be justified in the intro and your results and discussion may need to clearly split the two aims apart. I would like to see a clearer explanation of the rationale for your study. Why is it so important to understand women's experiences of the two different types of pain intervention? Methodology:
---

You need to clearly state the theoretical framework / qualitative approach used, and the data analysis method, and why these were chosen (I know you have explained your thematic analysis process, but this needs to link to the theoretical framework for the study). This is a vital component of the methodology of a qualitative study as the reader cannot be confident in your research approach nor your findings without this. Also, I would recommend adding a reference for the NVIVO software.

Did data analysis occur concurrently with data collection? Did you reach data saturation? If not, are these limitations of your study? What sort of questions were asked in the semi-structured interview? (perhaps an interview guide would be a useful appendix)

Results:

Overall I felt that the results could be presented in a way that better communicates your main findings. Perhaps an 'overview of findings' could be presented, which includes the name of each of the eight themes as well as a brief summary of what that theme is telling us about women's experiences of these two forms of pain intervention. I also think you may need to reconsider which of the eight themes are the most important in terms of meeting the aims of your research – and focus on those. This may also dictate the order of presentation of your themes. Your quotes are also quite long and numerous. I would consider being more selective and succinct in describing your themes. I think the readers would benefit greatly from a different approach to the presentation and order of your results.

Again, I think the 'impact on infant behaviour and feeding' doesn't currently integrate well with the other findings – being such a distinct topic it seems odd that only four quotes have been used to illustrate the theme (as opposed to 6+ for each of the others).

Discussion:

At present, I think the discussion lacks depth in exploring the MEANING and IPLICATIONS of the emergent themes, and integrating them with existing literature. Here is a reference that you may find useful to understand the difference between simply reporting themes and providing a deeper analysis: Bazeley P. Analysing qualitative data: more than 'identifying themes'. The Malasian Journal of Qualitative Research. 2009;2(2):3-5.

Ultimately, the discussion should address questions such as: What is the meaning of the themes - what are they telling us? How do they integrate with existing knowledge (lit)? How does this now change the way we understand the phenomenon of interest? And finally, what are the implications of this new knowledge?

I think that two very significant themes were 'cognitive effects' and 'enabling a sense of control', and I think these really deserve to be unpacked in the discussion as these relate very closely to our current understanding of the pain experience. For example, the cognitive effects of these drugs on women's ability to stay focussed during their labour can influence their pain experience – you may like to read one of our studies that also explores this concept - Whitburn LY, Jones LE, Davey M-A, Small R. Women's experiences of labour pain and the role of the mind: an exploratory study. Midwifery. 2014;30(9):1029-35. Particularly significant was the short-acting nature of remifentanyl, which allowed women to regain cognitive 'clarity' when needed, as opposed to pethidine. Also, the sense of 'control' over their pain created by the PCA mode of delivery of remifentanyl, as compared to pethidine, is really notable.

Limitations of your study have not been addressed.

	Abstract: See above for comment on the objectives. In the 'design' section, you should mention the data collection method as well as analysis method here (currently in 'setting'). Your results could be much more concise for the abstract. Results and conclusion may need a revision – see comments above. There were a number of spelling and formatting errors noted throughout – I have noted a few below but I suggest that each author undertakes another proof read to identify and correct any others. Page 2 (abstract), line 14-16 – check for inconsistent use of capitals Page 2 (abstract), line 51 – a trial that explored... Page 3, line 54 – “on demand” Page 5, line 44 – your referencing style is inconsistent with the rest of the manuscript here (Wilson et al 2018) Page 6, line 33 – this quote is in italics and quotation marks, whilst the rest are not Page 14, line 22 – Fleet and colleagues...
--	---

REVIEWER	Ban Leong Sng KK Women's and Children's Hospital, Singapore
REVIEW RETURNED	04-Aug-2019

GENERAL COMMENTS	The authors presented a qualitative study on women who received intramuscular pethidine or remifentanyl PCA for labour pain which is part of the RESPITE study. There is a formalised qualitative methodology that is described in detail. There are some suggestions: P3L3. Epidural analgesia has been shown to be the most effective form of pain relief for labour, but may give rise to unwanted side effects. Will be important to also state the rate of epidural uptake in the centers concerned due to patient choice and concerns and hence there is a need for alternative analgesic techniques such as labour opioid analgesia. P4L17. focus of this follow-up sub-study was to explore women's birth experiences, perceptions of their pain relief and infant feeding behaviours up to six weeks postpartum in women allocated to the pethidine, or remifentanyl trial arms for labour pain. Please be specific on the qualitative nature of this sub-study and what is the clinical equipoise for investigating using the qualitative methodology? P4L47. Women were recruited over a 16 month period and maternity units that recruited the most women within a one-month period received a £100 Love to Shop voucher. Would like to clarify if the remuneration was eventually given to how many units in the sub-study. P5L6. Thematic analysis of the entire data set was undertaken using Braun and Clark's approach¹¹. Would be good to clarify and give a summary of the principles of thematic analysis for the benefit of readers who may not be familiar with qualitative analyses. P5L29. Eighty trial participants from seven hospitals agreed to take part in the RESPITE Postnatal sub-study. Despite numerous contact attempts, thirty-one women were not interviewed due to women not responding or declining to be interviewed. Please use a study workflow to account for all the subjects, including the
--

	number of subjects from each of the 2 investigational groups that were recruited and the drop out from the various reasons. P6L3. Often women's approaches to labour were related to their previous childbirth experience. How many women were nulliparous? The narratives of those with previous childbirth and without previous childbirth could be different and the authors should comment on this. P7L31. Several women in the pethidine arm reported experiencing vomiting or nausea, How many women reported experiencing vomiting and nausea in this sub-study? P8L26. only a few women found remifentanil ineffective as pain relief. How many women found remifentanil ineffective, as would be important to know the qualitative measure out of the 30 subjects on remifentanil who were recruited. P8L36. Some women described experiencing 'no pain' after its administration, whereas others referred to how it made the pain more bearable: How many women found remifentanil useful, as would be important to know the qualitative measure out of the 30 subjects on remifentanil who were recruited. In general, an overall idea of the extend of the qualitative comments whether it is in the minority or majority of respondents will be helpful to estimate the extent of the problem. It is interesting that the study has revealed many aspects of analgesia and perinatal care that anaesthetists in general would not have known. P14L24. This study found that women who received fentanyl provided more positive, and less negative responses when compared to those who received pethidine9 Please clarify on the investigational product whether this should be remifentanil. Would be good to provide the number of positive responses compared to negative responses amongst the respondents. P14L51. It is important to note that these technical difficulties are not inevitable components of PCA, and are amenable to resolution. This would be a good learning point on the administrative points in PCA delivery and the authors could also give some suggestions or solutions to the technical difficulties reported in this study. Were there any comments from the pethidine group about ease of administration and advantages of single administration? This would be interesting to contrast that of PCA remifentanil. The discussion could be improved by information on the extent and whether a majority of women have the same responses or similar responses. P16L8. While these variations may be attributable to different numbers of women included from the trial groups and the level of postnatal support for breastfeeding across the different geographical areas, it may be that women who used remifentanil had fewer breastfeeding difficulties. The authors could include postulated mechanisms of why remifentanil could have less breastfeeding difficulties.
--	--

REVIEWER	Elisabetta Colciago University of Milano - Bicocca
REVIEW RETURNED	10-Aug-2019

GENERAL COMMENTS	Review bmjopen-2019-032203: A qualitative exploration of women's experiences of intramuscular pethidine or remifentanil patient controlled analgesia for labour pain
--

	Thank you for the opportunity to review the manuscript. The study has important clinical implications for both midwifery clinical practice and for women using these pain relief in labour. Please see below for specific comments on the manuscript. Abstract: Conclusions: The authors report in the discussion that this is the first study of its kind to be undertaken – this is important and worth including in the abstract. Introduction The authors reported that there is increasing interest in the role of remifentanil as an alternative opioid analgesia in Labour, and they presented benefits associated with it. They should also report the side effects associated with remifentanil, as was done for the other pain relievers before it. The aim of this study was to explore midwives' experience of the management of second stage of labour in women with epidural analgesia. No evidence have been reported regarding the breastfeeding intention and behaviour. Please include mention of any literature on this topic Methods It would be useful to provide some further descriptive information about the settings/Maternity Units of the study – i.e.: level of Maternities. Please include information on participants – inclusion criteria. Regarding the data analysis: Please clarify the role of the research team in the data analysis process. You mention that two midwives undertaking the interviews work through an iterative process, however is not clear how the research team was involved in agreeing on the final themes. Reference needed for NVIVO. The paper suggests that NVivo was used to analyse the data, although NVivo can support with the organisation and coding of data it does not in itself analyse data - this should be made clearer in the paper. Findings This section is well written. The use of quotes nicely illustrates the women's experiences. Discussion Well presented. You reported that a number of women in the remifentanil group reflected on how they were confused or altered, from quotes this side effect seemed quite strong. Please refer more explicitly to them - are your findings consistent/ in conflict with the existing research?
--	---

	- I wasn't feeling any pain at that point but it meant that I was almost removed from the labour process. (58R) - I must admit within, literally within I've no idea because I was out of it. [...] So then, again I don't know how long I was on it for. [...] Or maybe she was crying. I can't remember. I can't remember, sadly. (44R) Interviews undertook by telephone could be less effective than face-to-face ones, please report in the paper this weakness and the reasons why they are different. Page 15, line 38: I would change "opioid use during pregnancy" with "opioid use during labour" This paper should be considered for publication following satisfactory revision.
--	---

REVIEWER	Dr M Anim-Somuah Tameside and Glossop Integrated Care NHS Foundation Trust Uk
REVIEW RETURNED	12-Aug-2019

GENERAL COMMENTS	Good points Good piece of qualitative research with a clear purpose of exploring women's birth experiences of pain relief and infant feeding intentions and behaviours up to 6 weeks postpartum in women who received Remifentanyl or Pethidine for labour pain. Relevant question being addressed as no previous qualitative research addressing women's experiences of infant feeding intentions of women who used IV Remifentanyl for pain relief in labour. Clear pre-specified qualitative exploratory follow up sub study of RESPITE trial Methodology clearly defined with appropriate use of thematic analysis following semi structured interviews Maintained confidentiality of participants Relevant results drawn from analysis and excellently presented in manuscript Point that could be improved (Minor points) Prior definition of birth experiences in study Exploration/ possible explanation of why fewer women in Pethidine arm participated in sub study and how this may have affected results Major point Conclusions in the main text needs to be reflected in abstract. Results under Abstract Page 2, 45-47 'there appears to be little difference reported in breastfeeding.... continuation rates' and Abstract conclusion Page 2, 51-53 'insights from follow up study that explored experiences of intravenous remifentanyl PCA and intra-muscular pethidine injection yielded little difference in breastfeeding experiences between groups. However Conclusions in main text Page 16, 6-8 '... there was an indication that women who received remifentanyl were more likely to continue breastfeeding...'
--

REVIEWER	Cynthia A. Wong University of Iowa, USA
REVIEW RETURNED	12-Aug-2019

GENERAL COMMENTS	BMJ Open 2019-032203 General Comments: The manuscript describes a substudy of the RESPITE trial (a multicenter, open label, RCT comparing remifentanil PCA to IM, midwife-administered pethidine for labor analgesia). The substudy enrolled a subset of these patients to receive a 6-week, birth experience related to pain control/satisfaction, and possible adverse effects. The questions the study wished to address are valid and of interest to clinicians providing labor analgesia. However, I have some significant concerns about the study design (see below), and thus the conclusions that can be drawn from the current study. Major Concerns:  1. A limitation of the study is that a small subset of patients who participated in the original study enrolled in the current study (N = 400 in original study, N = 49 in current study). The authors have not presented data to support that this small group is representative of the entire group (see Comment 10). 2. A major limitation of the current study is likely selection bias. It is very possible that women who were dissatisfied with pethidine agreed to participate in the study at a higher rate than women who were dissatisfied with remifentanil, and women who were happy with remifentanil agree to participate at a higher rate than those who were unhappy. I am unclear why many of the questions posed in the interview, particularly the questions about pain control and side effects, were not asked in the immediate postpartum period when the study subjects were still inpatients. This would likely have resulted in many more women participating and a lower risk for selection bias. Because of this risk of selection bias, I suggest that the authors should not/cannot state “most women in the remifentanil (pethidine) group” or “more women in the remifentanil compared to pethidine group....” All similar statements should be removed from the manuscript. 3. The planned substudy is not mentioned in the original manuscript. It is listed as a secondary outcome in the study registry (“8. Explore and compare women’s birth experiences, perceptions of pain relief and infant feeding behaviours up to six weeks postpartum (RESPITE Post-Natal Sub-Study)”; however, it is not clear when the substudy was conceived relative to the original study. The following editorial note is included in the study registration: 21/12/2015: At the trialist's request, the record has undergone substantial updates. “The recruitment end date and overall trial end date have been extended from 30/11/2015 to 30/09/2016. The secondary outcomes and participant exclusion criteria have been updated.” (application date 8/8/2013, patients enrollment for the RESPITE trial began in May 2014). Specific Comments:  1. P3, Strengths: Please do not claim to be “the first” unless a literature review strategy has been described to rule out the possibility that someone else has done this previously. 2. P3, Strengths: I do not agree that 49 is “large.”
---

3. P3, Strengths and other places in the manuscript: The word “epidural” is an adjective, and should be followed by a noun, e.g., “epidural analgesia.”
4. P3L38: The data to support this statement (“pethidine is the most common opioid for labor analgesia in the UK” is well over a decade old (published in 2009, included studies as old as the 1970’s). Are there more recent data? It is certainly no longer true in the US that pethidine is the most commonly used opioid for labor analgesia; this has not been the case for quite some time.
5. P3L43-5: This sentence implies that lack of mobilization and its associated prolonged and painful labor are characteristics that are unique to pethidine analgesia. To my knowledge, these associations are true, no matter the type of analgesia, or whether analgesia is used at all.
6. P3L45-8: More accurately, I believe these outcomes have been associated with use of pethidine analgesia, but I don’t believe there is conclusion of causality. Women who chose labor analgesia, no matter the type, have difference labor and postpartum characteristics than women who do not.
7. P4L39: Which 7 hospitals participated and why only 7 of the original 14?
8. P5, Sample size: Please justify the sample size. It appears to be a convenience sample.
9. P5, Results: Please note the inclusive dates of the study.
10. P5L35: I suggest the authors compare baseline and delivery characteristics between women who agreed to participate to those who did not. Did these groups differ?
11. P5L35-9: The administration of pethidine before randomization appears to be a study protocol violation. Why was this not reported in the original publication?
12. P5L39-40: Again, I am concerned that the use of nitrous oxide was not reported in the original publication. Nitrous oxide may contribute to side effects, including sedation, which was an outcome of the study.
13. P5L44-6: Did any women who did not receive epidural analgesia have a cesarean delivery?
14. P5L54: It is not clear to me why the study subjects numbers are revealed. This is also confusing, because it implies that just the one patient mentioned this theme (e.g., “there weren’t any plans..” I am sure that more than one person had no plan. Please delete the numbers.
15. P7L31-3: Please revise sentence so that it does not imply that nausea was more frequent in the pethidine group compared to the remifentanil group. This cannot be determined from the current study design (see Major Comment 2 above).
16. P6-7, Negative Physiologic Responses and P10-12, Cognitive Effects: I am unclear how the authors differentiated some of the negative physiologic responses from the cognitive effects. There appears to be a great deal of overlap between some negative physiologic effects and cognitive effects.
17. P14-15, Discussion: A large part of the discussion is repetition of the results section. This section could be shorter without losing important content.
18. P15L18-30: Please do not imply that there are differences in the rates of breastfeeding between the two groups. This cannot be determined from the current study design (see Major Comment 2).
19. P5L55-7, P16L6-8: Again, please do not compare the two study groups—the study has a high risk of selection bias (see Major Comment#2).

	20. Table 1: Parity should be “nulliparous” and “parous” (the parous group also includes primiparous, which is not multiparous). Respectfully reviewed, Cynthia A. Wong, MD
--	--

VERSION 1 – AUTHOR RESPONSE

Reviewer(s)' Comments to Author:

Reviewer: 1

Reviewer Name: Dr Laura Whitburn

Institution and Country: La Trobe University, Victoria, Australia

Please state any competing interests or state 'None declared': None declared

Please leave your comments for the authors below

Thank you for giving me the opportunity to review this manuscript. Women's experiences of labour pain and pain interventions is of great importance and the findings can contribute towards the care of women through labour and birth.

This manuscript has the beginnings of a valuable paper, but I feel that it still needs some refinement, particularly regarding the presentation of the results and the discussion.

Intro:

The first paragraph of the intro could better focus the topic of the paper. I think that the first sentence, whilst VERY important, seems a bit out of place as it is not well integrated with the next. You may wish to work on this.

Thank you for your suggestion. We have refocused the opening sentence of the manuscript.

The objective may also need to be reconsidered / rewritten. As it currently reads (in the abstract and in introduction), it sounds like you are exploring women's birth experiences of pain relief and infant feeding intentions and behaviours – however of course infant feeding intentions and behaviours are pre- and post-natal experiences respectively. I think that the primary aim was to explore women's birth experiences of the two different forms of pain relief. However, the aim to explore women's infant feeding intentions and behaviours does not quite fit – there is no mention of the relationship between pethidine/remifentanil and breastfeeding at all in the introduction. You may need to make the decision to focus only on the pain experience, or, if you intend to include the data on feeding, this should be justified in the intro and your results and discussion may need to clearly split the two aims apart.

Thank you for your comment, however we maintain that one of our a priori aims was to explore infant feeding behaviours following administration of remifentanil/pethidine. We also collected data on mothers' infant feeding intentions. However we agree that the inclusion of infant feeding intentions (which occurs prenatally) in our aims is confusing and have deleted this from our aims (abstract and

introduction, p 5). We report infant feeding intentions in our results to describe the characteristics of our sample.

As well as the ethical issue of ensuring we fully represent our participant's views, we believe it is important to retain our findings on infant feeding behaviours as this is the first study to have explored this qualitatively. We have added an additional sentence in our introduction (p4) about the lack of research investigating remifentanyl and infant feeding behaviours (there is already a sentence relating to pethidine and infant feeding behaviours).

I would like to see a clearer explanation of the rationale for your study. Why is it so important to understand women's experiences of the two different types of pain intervention?

We have included a rationale for our study (P.4)

Methodology:

You need to clearly state the theoretical framework / qualitative approach used, and the data analysis method, and why these were chosen (I know you have explained your thematic analysis process, but this needs to link to the theoretical framework for the study). This is a vital component of the methodology of a qualitative study as the reader cannot be confident in your research approach nor your findings without this. Also, I would recommend adding a reference for the NVIVO software.

We have stated on P4 (Design) that this is a qualitative exploratory study. This was an appropriate approach to take given the limited research in this area, particularly with regard to remifentanyl.

The reference for NVIVO has been added to our reference list.

Did data analysis occur concurrently with data collection? Did you reach data saturation? If not, are these limitations of your study?

We have added that data analysis occurred concurrently with data collection on p5. There is much debate about the value of data saturation. We did, however, interview as many women as possible and did not find any new themes occurring from our data.

What sort of questions were asked in the semi-structured interview? (perhaps an interview guide would be a useful appendix)

We have added an example of questions asked and the interview guide can be made available at the Editor's request.

Results:

Overall I felt that the results could be presented in a way that better communicates your main findings. Perhaps an 'overview of findings' could be presented, which includes the name of each of the eight themes as well as a brief summary of what that theme is telling us about women's experiences of these two forms of pain intervention. I also think you may need to reconsider which of the eight themes are the most important in terms of meeting the aims of your research – and focus on those.

This may also dictate the order of presentation of your themes. Your quotes are also quite long and numerous. I would consider being more selective and succinct in describing your themes. I think the readers would benefit greatly from a different approach to the presentation and order of your results.

We have now provided an overview of our findings on P7-8.

We feel it would be inappropriate to super-value any particular theme and unethical if we did not present all the issues that were raised by our participants. We have added that all themes were interlinking (P7).

We have edited some of the quotes, however we do feel it is important to provide quotes from women who received both pethidine and remifentanyl.

Again, I think the 'impact on infant behaviour and feeding' doesn't currently integrate well with the other findings – being such a distinct topic it seems odd that only four quotes have been used to illustrate the theme (as opposed to 6+ for each of the others).

As discussed earlier, the impact of remifentanyl on breastfeeding has not previously been explored in qualitative research (and only briefly in the associated RESPITE RCT). We feel that it is important to keep this data as opioid use has previously been associated with breastfeeding problems. We acknowledge that it is tentative data, but the finding that no similar problems were identified with remifentanyl is important.

Discussion:

At present, I think the discussion lacks depth in exploring the MEANING and IMPLICATIONS of the emergent themes, and integrating them with existing literature. Here is a reference that you may find useful to understand the difference between simply reporting themes and providing a deeper analysis: Bazeley P. Analysing qualitative data: more than 'identifying themes'. *The Malaysian Journal of Qualitative Research*. 2009;2(2):3-5.

Ultimately, the discussion should address questions such as: What is the meaning of the themes - what are they telling us? How do they integrate with existing knowledge (lit)? How does this now change the way we understand the phenomenon of interest? And finally, what are the implications of this new knowledge?

I think that two very significant themes were 'cognitive effects' and 'enabling a sense of control', and I think these really deserve to be unpacked in the discussion as these relate very closely to our current understanding of the pain experience. For example, the cognitive effects of these drugs on women's ability to stay focussed during their labour can influence their pain experience – you may like to read one of our studies that also explores this concept - Whitburn LY, Jones LE, Davey M-A, Small R. Women's experiences of labour pain and the role of the mind: an exploratory study. *Midwifery*. 2014;30(9):1029-35. Particularly significant was the short-acting nature of remifentanyl, which allowed women to regain cognitive 'clarity' when needed, as opposed to pethidine. Also, the sense of 'control' over their pain created by the PCA mode of delivery of remifentanyl, as compared to pethidine, is really notable.

Thank you for your suggestions, we have added to our discussion.

Limitations of your study have not been addressed.

We discussed the limitations of our study on page 18.

Abstract:

See above for comment on the objectives. In the 'design' section, you should mention the data collection method as well as analysis method here (currently in 'setting'). Your results could be much more concise for the abstract. Results and conclusion may need a revision – see comments above.

Thank you for your comment. This has been amended.

There were a number of spelling and formatting errors noted throughout – I have noted a few below but I suggest that each author undertakes another proof read to identify and correct any others.

Page 2 (abstract), line 14-16 – check for inconsistent use of capitals

Thank you for your comment. This has been amended.

Page 2 (abstract), line 51 – a trial that explored...

Thank you for your comment. This has been amended.

Page 3, line 54 – “on demand”

Thank you for your comment. This has been amended.

Page 5, line 44 – your referencing style is inconsistent with the rest of the manuscript here (Wilson et al 2018)

Thank you for your comment. This has been amended.

Page 6, line 33 – this quote is in italics and quotation marks, whilst the rest are not

Thank you for your comment. This has been amended.

Page 14, line 22 – Fleet and colleagues...

Thank you for your comment. This has been amended.

Reviewer: 2

Reviewer Name: Ban Leong Sng

Institution and Country: KK Women's and Children's Hospital, Singapore

Please state any competing interests or state 'None declared': Non declared

Please leave your comments for the authors below

The authors presented a qualitative study on women who received intramuscular pethidine or remifentanyl PCA for labour pain which is part of the RESPITE study. There is a formalised qualitative methodology that is described in detail. There are some suggestions:

P3L3. Epidural analgesia has been shown to be the most effective form of pain relief for labour, but may give rise to unwanted side effects. Will be important to also state the rate of epidural uptake in the centers concerned due to patient choice and concerns and hence there is a need for alternative analgesic techniques such as labour opioid analgesia.

Unfortunately, despite contacting all seven centres, only two replied so we were unable to include this information in our manuscript.

P4L17. focus of this follow-up sub-study was to explore women's birth experiences, perceptions of their pain relief and infant feeding behaviours up to six weeks postpartum in women allocated to the pethidine, or remifentanyl trial arms for labour pain. Please be specific on the qualitative nature of this sub-study and what is the clinical equipoise for investigating using the qualitative methodology?

We have added a rationale for our study, as also suggested by Reviewer 1, on page 4.

P4L47. Women were recruited over a 16 month period and maternity units that recruited the most women within a one-month period received a £100 Love to Shop voucher. Would like to clarify if the remuneration was eventually given to how many units in the sub-study.

£100 vouchers were sent out to 4 sites. We have clarified this in the manuscript (p6)

P5L6. Thematic analysis of the entire data set was undertaken using Braun and Clark's approach¹¹. Would be good to clarify and give a summary of the principles of thematic analysis for the benefit of readers who may not be familiar with qualitative analyses.

P5 we have inserted additional text to highlight that thematic analysis was used to identify patterned meaning across the dataset.

P5L29. Eighty trial participants from seven hospitals agreed to take part in the RESPITE Postnatal sub-study. Despite numerous contact attempts, thirty-one women were not interviewed due to women not responding or declining to be interviewed. Please use a study workflow to account for all the subjects, including the number of subjects from each of the 2 investigational groups that were recruited and the drop out from the various reasons.

Thank you for your comment. Of the 31 women who did not take part after initial consent, the majority (n=23) were removed from the study by the research team due to being unavailable in the study timeframe or due to language barriers. The remaining 8 women chose to withdraw from the study.

This is clarified on P6. We did not feel it appropriate to seek reasons for their withdrawal as our PIS stated that women were free to withdraw from the study at any point. We did not think it was necessary to provide a workflow diagram, but will be guided by the editors.

P6L3. Often women's approaches to labour were related to their previous childbirth experience. How many women were nulliparous? The narratives of those with previous childbirth and without previous childbirth could be different and the authors should comment on this.

The majority of women were nulliparous. This information is given in Table 1. We have added the following to the text: As no stark differences were found between the responses of nulliparous and parous women no differentiation was made in the findings. (P7)

P7L31. Several women in the pethidine arm reported experiencing vomiting or nausea, How many women reported experiencing vomiting and nausea in this sub-study?

P8L26. only a few women found remifentanil ineffective as pain relief. How many women found remifentanil ineffective, as would be important to know the qualitative measure out of the 30 subjects on remifentanil who were recruited.

P8L36. Some women described experiencing 'no pain' after its administration, whereas others referred to how it made the pain more bearable: How many women found remifentanil useful, as would be important to know the qualitative measure out of the 30 subjects on remifentanil who were recruited.

In general, an overall idea of the extent of the qualitative comments whether it is in the minority or majority of respondents will be helpful to estimate the extent of the problem. It is interesting that the study has revealed many aspects of analgesia and perinatal care that anaesthetists in general would not have known.

P14L24. This study found that women who received fentanyl provided more positive, and less negative responses when compared to those who received pethidine. Please clarify on the investigational product whether this should be remifentanil. Would be good to provide the number of positive responses compared to negative responses amongst the respondents.

P14L51. It is important to note that these technical difficulties are not inevitable components of PCA, and are amenable to resolution. This would be a good learning point on the administrative points in PCA delivery and the authors could also give some suggestions or solutions to the technical difficulties reported in this study. Were there any comments from the pethidine group about ease of administration and advantages of single administration? This would be interesting to contrast that of PCA remifentanil.

The discussion could be improved by information on the extent and whether a majority of women have the same responses or similar responses.

We thank you for these comments, however this study was not designed to use content analysis (which would be more appropriate with a structured, rather than semi-structured interview schedule). We feel it would be inappropriate to make quantitative comparisons between groups within the context of a qualitative study.

P16L8. While these variations may be attributable to different numbers of women included from the trial groups and the level of postnatal support for breastfeeding across the different geographical

areas, it may be that women who used remifentanil had fewer breastfeeding difficulties. The authors could include postulated mechanisms of why remifentanil could have less breastfeeding difficulties.

We have added the following text to P18: These differences may relate to the differential effects of analgesia on infant behaviours which may impact of establishment of sustained breastfeeding. However, as there are many biopsychosocial factors that can influence the mother's infant feeding decisions, further research is needed.

Reviewer: 3

Reviewer Name: Elisabetta Colciago

Institution and Country: University of Milano - Bicocca

Please state any competing interests or state 'None declared': Non Declared

Please leave your comments for the authors below

Review bmjopen-2019-032203: A qualitative exploration of women's experiences of intramuscular pethidine or remifentanil patient controlled analgesia for labour pain

Thank you for the opportunity to review the manuscript.

The study has important clinical implications for both midwifery clinical practice and for women using these pain relief in labour.

Please see below for specific comments on the manuscript.

Abstract: Conclusions: The authors report in the discussion that this is the first study of its kind to be undertaken – this is important and worth including in the abstract.

Unfortunately word restrictions on the abstract negates the inclusion of this point

Introduction

The authors reported that there is increasing interest in the role of remifentanil as an alternative opioid analgesia in Labour, and they presented benefits associated with it. They should also report the side effects associated with remifentanil, as was done for the other pain relievers before it.

Thank you for your comment. Information on side effects associated with remifentanil has been added on page 4.

The aim of this study was to explore midwives' experience of the management of second stage of labour in women with epidural analgesia. No evidence have been reported regarding the breastfeeding intention and behaviour.

Please include mention of any literature on this topic

The reviewer is mistaken. The aim of the study was to explore women's experiences of opioid use for labour pain. We did not seek midwives' experiences, nor did we focus on epidural use. We have added a sentence which states that the influence of remifentanil on infant feeding behaviours remains largely unexplored.

Methods

It would be useful to provide some further descriptive information about the settings/Maternity Units of the study – i.e.: level of Maternities.

The maternity units were all obstetric led and did not routinely use remifentanil PCA prior to being involved with the trial. Additional detail is added to P5.

Please include information on participants – inclusion criteria.

Thank you for your comment. Information on inclusion criteria has been added to page 5.

Regarding the data analysis: Please clarify the role of the research team in the data analysis process. You mention that two midwives undertaking the interviews work through an iterative process, however is not clear how the research team was involved in agreeing on the final themes.

The interviewers were research assistants, not midwives. We have clarified the role of two of the other authors (VHM, GT) in the data analysis

Reference needed for NVIVO

This has been inserted into the reference list.

The paper suggests that NVivo was used to analyse the data, although NVivo can support with the organisation and coding of data it does not in itself analyse data - this should be made clearer in the paper.

We have amended our text to clarify the role of NVivo (P5).

Findings

This section is well written. The use of quotes nicely illustrates the women's experiences.

Discussion

Well presented.

You reported that a number of women in the remifentanyl group reflected on how they were confused or altered, from quotes this side effect seemed quite strong. Please refer more explicitly to them - are your findings consistent/ in conflict with the existing research?

- I wasn't feeling any pain at that point but it meant that I was almost removed from the labour process. (58R)

- I must admit within, literally within I've no idea because I was out of it. [...] So then, again I don't know how long I was on it for. [...] Or maybe she was crying. I can't remember. I can't remember, sadly. (44R)

We have compared these cognitive effects with existing research on page 17 (also see our response to reviewer 1)

Interviews undertaken by telephone could be less effective than face-to-face ones, please report in the paper this weakness and the reasons why they are different.

There is little evidence to support the superiority of face to face interviewing over telephone interviews nor is there evidence that the telephone interviews produce lower quality data. Whilst it is acknowledged that the absence of visual cues via telephone may result in loss of contextual and nonverbal data and to compromise rapport, probing, and interpretation of responses, telephone interviewing may allow respondents to feel relaxed and able to disclose sensitive information. Indeed research comparing the two modes of interview have shown very little difference between them (Vogl 2013). In addition, telephone interviews may be well-suited to potentially sensitive topics because this technique provides participants with the opportunity to disclose intimate and closely held experiences without feeling uncomfortable (Sturges & Hanrahan 2004).

We have added the following text to page 5: Telephone interviews provided participants with the opportunity to disclose intimate and sensitive experiences without feeling uncomfortable.

Vogl, S., 2013. Telephone versus face-to-face interviews: Mode effect on semistructured interviews with children. *Sociological Methodology*, 43(1), pp.133-177.

Sturges, J. E., & Hanrahan, K. J. (2004). Comparing Telephone and Face-to-Face Qualitative Interviewing: A Research Note. *Qualitative Research*, 4, 107-118.

Page 15, line 38: I would change "opioid use during pregnancy" with "opioid use during labour"

Thank you for your comment. This has been amended.

This paper should be considered for publication following satisfactory revision.

Reviewer: 4

Reviewer Name: Dr M Anim-Somuah

Institution and Country: Tameside and Glossop Integrated Care NHS Foundation Trust

Uk

Please state any competing interests or state 'None declared': None Declared

Please leave your comments for the authors below

Good points

Good piece of qualitative research with a clear purpose of exploring women's birth experiences of pain relief and infant feeding intentions and behaviours up to 6 weeks postpartum in women who received Remifentanyl or Pethidine for labour pain.

Relevant question being addressed as no previous qualitative research addressing women's experiences of infant feeding intentions of women who used IV Remifentanyl for pain relief in labour.

Clear pre-specified qualitative exploratory follow up sub study of RESPITE trial

Methodology clearly defined with appropriate use of thematic analysis following semi structured interviews

Maintained confidentiality of participants

Relevant results drawn from analysis and excellently presented in manuscript

Point that could be improved (Minor points)

Prior definition of birth experiences in study

We have now clarified how many women were parous and nulliparous.

Exploration/ possible explanation of why fewer women in Pethidine arm participated in sub study and how this may have affected results

Thank you for your comment, however, we feel that an exploration of why fewer women participated in the pethidine arm is beyond the scope of our qualitative design. We have already cited the smaller number of women in the pethidine arm as a potential limitation of our study.

Major point

Conclusions in the main text needs to be reflected in abstract.

Results under Abstract Page 2, 45-47 'there appears to be little difference reported in breastfeeding... continuation rates' and Abstract conclusion Page 2, 51-53 'insights from follow up study that explored experiences of intravenous remifentanil PCA and intra-muscular pethidine injection yielded little difference in breastfeeding experiences between groups. However Conclusions in main text Page 16, 6-8 '... there was an indication that women who received remifentanil were more likely to continue breastfeeding...'.

Thank you for your comment. This has been amended in the abstract.

Reviewer: 5

Reviewer Name: Cynthia A. Wong

Institution and Country: University of Iowa, USA

Please state any competing interests or state 'None declared': I have no competing interests.

Please leave your comments for the authors below

BMJ Open 2019-032203

General Comments: The manuscript describes a substudy of the RESPITE trial (a multicenter, open label, RCT comparing remifentanil PCA to IM, midwife-administered pethidine for labor analgesia). The substudy enrolled a subset of these patients to receive a 6-week, birth experience related to pain control/satisfaction, and possible adverse effects. The questions the study wished to address are valid and of interest to clinicians providing labor analgesia. However, I have some significant concerns about the study design (see below), and thus the conclusions that can be drawn from the current study.

Major Concerns:

1. A limitation of the study is that a small subset of patients who participated in the original study enrolled in the current study (N = 400 in original study, N = 49 in current study). The authors have not presented data to support that this small group is representative of the entire group (see Comment 10).

We have acknowledged the smaller number of pethidine participants as compared to remifentanil participants as a limitation of the study (P18) but also maintain that an N of 49 is very large for a qualitative study. It is not the intention of qualitative research to provide generalisability.

2. A major limitation of the current study is likely selection bias. It is very possible that women who were dissatisfied with pethidine agreed to participate in the study at a higher rate than women who were dissatisfied with remifentanyl, and women who were happy with remifentanyl agree to participate at a higher rate than those who were unhappy. I am unclear why many of the questions posed in the interview, particularly the questions about pain control and side effects, were not asked in the immediate postpartum period when the study subjects were still inpatients. This would likely have resulted in many more women participating and a lower risk for selection bias. Because of this risk of selection bias, I suggest that the authors should not/cannot state “most women in the remifentanyl (pethidine) group” or “more women in the remifentanyl compared to pethidine group....” All similar statements should be removed from the manuscript.

We agree that the study may be affected by selection bias (i.e. that women who held stronger views towards the use of analgesia (positive or negative) took part in the study), as we have indicated in our limitations on P18. However we disagree that those who agreed to participate would be more likely to be dissatisfied with pethidine/satisfied with remifentanyl. The aim of the RESPITE trial was to compare intravenous remifentanyl PCA with intramuscular pethidine injection in labour – there were no underlying assumptions that remifentanyl would be superior and better received by the women taking this analgesic. Similarly with our qualitative study – we held no prior assumptions and so we’re unsure why the reviewer thinks that the women would feel this way.

It is standard practice to describe qualitative data using adjectives such as ‘more’ or ‘most’.

We interviewed women at 6 weeks following birth rather than in the immediate postpartum period because it has been suggested that women may respond more positively towards childbirth if interviewed/consulted in the early postnatal period – thereby reflecting the ‘halo’ effect. It has also been reported that emotional impacts of childbirth may only emerge sometime after the event (Crompton, 2003; Waldenstrom, 2003). Women may also be reticent in raising complaints about maternity care, particularly if being asked to make judgements whilst still in the maternity environment (Lumley, 1985). It has been suggested that a time interval should have elapsed prior to the assessment of maternal outcomes. The rationale is that this would allow the participants to gain an embedded perspective of their experience (Lumley, 1985; Bramadat & Driedger, 1993), and that their reflections are more contextualised.

- Bramadat, I.J., Driedger, M., 1993. Satisfaction with childbirth: theories and methods of measurement. *Birth*, 20 (1), 22-29.
- Crompton, J., 2003. Post-traumatic stress disorder and childbirth. *New Zealand College of Midwives Journal*, 26, 17-19.
- Lumley, J., 1985. Assessing satisfaction with childbirth. *Birth*, 12, 141-145.
- Waldenstrom, U., 2003. Why do some women change their opinion about childbirth over time? *Birth*, 31 (2), 102-107.

3. The planned substudy is not mentioned in the original manuscript. It is listed as a secondary outcome in the study registry (“8. Explore and compare women’s birth experiences, perceptions of

pain relief and infant feeding behaviours up to six weeks postpartum (RESPITE Post-Natal Sub-Study”); however, it is not clear when the substudy was conceived relative to the original study. The following editorial note is included in the study registration: 21/12/2015: At the trialist's request, the record has undergone substantial updates. “The recruitment end date and overall trial end date have been extended from 30/11/2015 to 30/09/2016. The secondary outcomes and participant exclusion criteria have been updated.” (application date 8/8/2013, patients enrollment for the RESPITE trial began in May 2014).

The sub-study was conducted primarily by researchers (based at the University of Central Lancashire) who were not directly involved with the RESPITE trial. We approached the RESPITE trial team after hearing about the trial and asked if we could conduct a qualitative study alongside the RCT, hence the amendment to the registry. It was originally intended that the sub-study would open to recruitment at the same time as the main trial but this was delayed due to having to get approval for the sub-study via a substantial amendment as well as having contracts etc in place. Recruitment to the main trial opened in May 2014 and the sub-study opened to recruitment in June 2015. The original RESPITE trial manuscript published in The Lancet deliberately only focuses on the results of the main trial as it was always intended to produce a separate manuscript for the sub-study. The trial protocol (which can be made available on request) includes an appendix which states in detail the aims and methods for the sub-study.

Specific Comments:

1. P3, Strengths: Please do not claim to be “the first” unless a literature review strategy has been described to rule out the possibility that someone else has done this previously.

Thank you for your comment. This has been amended.

2. P3, Strengths: I do not agree that 49 is “large.”

We state here that we conducted a ‘large number of interviews’. We stand by our claim that 49 interviews is a large number in the context of qualitative research, but have changed the wording to ‘Substantial number of women (49) interviewed..’ to avoid confusion.

3. P3, Strengths and other places in the manuscript: The word “epidural” is an adjective, and should be followed by a noun, e.g., “epidural analgesia.”

Thank you for your comment. This has been amended throughout.

4. P3L38: The data to support this statement (“pethidine is the most common opioid for labor analgesia in the UK” is well over a decade old (published in 2009, included studies as old as the 1970’s). Are there more recent data? It is certainly no longer true in the US that pethidine is the most commonly used opioid for labor analgesia; this has not been the case for quite some time.

We have altered the text slightly and updated the reference

5. P3L43-5: This sentence implies that lack of mobilization and its associated prolonged and painful labor are characteristics that are unique to pethidine analgesia. To my knowledge, these associations are true, no matter the type of analgesia, or whether analgesia is used at all.

Thank you for your comment. We have deleted this point.

6. P3L45-8: More accurately, I believe these outcomes have been associated with use of pethidine analgesia, but I don't believe there is conclusion of causality. Women who chose labor analgesia, no matter the type, have difference labor and postpartum characteristics than women who do not.

We have made this sentence more tentative, thereby avoiding suggestion of causality

7. P4L39: Which 7 hospitals participated and why only 7 of the original 14?

The following 7 hospitals recruited women to the sub-study:: Birmingham Women's Hospital, Royal Stoke University Hospital, University Hospital Coventry, Warwick Hospital, York Hospital, Bradford Royal Infirmary and Norfolk and Norwich University Hospital.

We have added the following text to the manuscript: Whilst all fourteen sites participating in the RESPITE trial were eligible for the sub-study, only seven hospitals successfully consented participants who progressed to interview.

However we do not have permission to name the participating trusts in the manuscript.

8. P5, Sample size: Please justify the sample size. It appears to be a convenience sample.

Our sample was a convenience sample. Sample sizes in qualitative research are highly contested, with different sample sizes suggested for different study designs, i.e. 30-50 for ethnographic studies (Morse, 1994) and 20-30 for grounded theory (Cresswell, 1998); with a minimum of 15 suggested for any type of qualitative research (Guest et al., 2006). According to Sandelowski (1995) determining adequate sample size in qualitative research is ultimately a matter of judgment and experience in evaluating the quality of the information collected against the uses to which it will be put.

- Morse, Janice M. (1994). Designing funded qualitative research. In Norman K. Denzin & Yvonna S. Lincoln (Eds.), *Handbook of qualitative research* (2nd ed., pp.220-35). Thousand Oaks, CA: Sage.
- Creswell, John (1998). *Qualitative inquiry and research design: Choosing among five traditions*. Thousand Oaks, CA: Sage.
- Guest, Greg; Bunce, Arwen & Johnson, Laura (2006). "How many interviews are enough? An

- experiment with data saturation and variability". *Field Methods*, 18(1), 59-82.
- Sandelowski, M. (1995). Sample size in qualitative research. *Research in Nursing & Health*, 18(2), 179-183

9. P5, Results: Please note the inclusive dates of the study.

Women were consented for the sub-study at sites between June 2015 and September 2016. We have added this information on P 5.

10. P5L35: I suggest the authors compare baseline and delivery characteristics between women who agreed to participate to those who did not. Did these groups differ?

We feel that this is beyond the scope and inappropriate for our qualitative study.

11. P5L35-9: The administration of pethidine before randomization appears to be a study protocol violation. Why was this not reported in the original publication?

Women were allowed to have received previous systemic opioid pain relief if it was more than 4 hours prior to randomisation. This is stated on P6

12. P5L39-40: Again, I am concerned that the use of nitrous oxide was not reported in the original publication. Nitrous oxide may contribute to side effects, including sedation, which was an outcome of the study.

The use of nitrous oxide was not a primary or secondary outcome for the main trial, which is why it was not included in the main publication. However whether the woman used Entonox ("gas and air") at any time after study entry was recorded on the 'Pain Relief and Maternal Observations' form.

In the main trial, of the 201 women randomised to remifentanyl 149 (74%) used Entonox. Of the 199 women randomised to pethidine 179 women (90%) used Entonox.

13. P5L44-6: Did any women who did not receive epidural analgesia have a cesarean delivery?

8/49 women had a caesarean delivery and of these 3 did not have an epidural. This is clarified on P6

14. P5L54: It is not clear to me why the study subjects numbers are revealed. This is also confusing, because it implies that just the one patient mentioned this theme (e.g., "there weren't any plans..") I am sure that more than one person had no plan. Please delete the numbers.

It is common place to use identifiers for quotations, the numbers, followed by R or P to denote the type of opioid, are necessary to illustrate that the quotes originate from a range of participants.

15. P7L31-3: Please revise sentence so that it does not imply that nausea was more frequent in the pethidine group compared to the remifentanil group. This cannot be determined from the current study design (see Major Comment 2 above).

The theme of nausea came up far more frequently in the pethidine group. It is commonplace to use adjectives such as 'several', 'more' etc when describing qualitative data.

16. P6-7, Negative Physiologic Responses and P10-12, Cognitive Effects: I am unclear how the authors differentiated some of the negative physiologic responses from the cognitive effects. There appears to be a great deal of overlap between some negative physiologic effects and cognitive effects..

We have edited the themes to avoid repetition. Physiological response themes include drowsiness, lack of mobility, vomiting, and nausea. Whereas cognitive themes include a sense of vulnerability, feeling out of it (like feeling intoxicated), confusion, impacting on memory. We have also now made it explicit that the themes were interlinking (see p. 7).

17. P14-15, Discussion: A large part of the discussion is repetition of the results section. This section could be shorter without losing important content.

Thank you for your comment, we have made some edits.

18. P15L18-30: Please do not imply that there are differences in the rates of breastfeeding between the two groups. This cannot be determined from the current study design (see Major Comment 2).

We have been very tentative about this suggestion, and we do not imply any causality.

19. P5L55-7, P16L6-8: Again, please do not compare the two study groups—the study has a high risk of selection bias (see Major Comment#2).

See our response to #2

20. Table 1: Parity should be “nulliparous” and “parous” (the parous group also includes primiparous, which is not multiparous).

Thank you for your comment. This has been amended.

Respectfully reviewed,

Cynthia A. Wong, MD

VERSION 2 – REVIEW

REVIEWER	Dr Laura Whitburn La Trobe University, Victoria, Australia
REVIEW RETURNED	25-Oct-2019

GENERAL COMMENTS	Thank you for the opportunity to review this revised manuscript. The authors have addressed all points raised and in my opinion the manuscript reads very well. I have two minor points that should be corrected prior to publication: P16L52: In reference to previous research done by Whitburn and colleagues, you have mis-typed the findings causing a significant error in the reported results from this study: it should read 'They found that an inability...' I also noted numerous formatting and grammatical issues throughout the manuscript - see file attached. Well done to the authors on this very important research. The reviewer provided a marked copy with additional comments. Please contact the publisher for full details.
--

REVIEWER	Millicent Anim-Somuah Tameside and Glossop Integrated Care NHS Foundation Trust United Kingdom
REVIEW RETURNED	08-Nov-2019

GENERAL COMMENTS	Review comments enacted upon. Recommend for publication
--

VERSION 2 – AUTHOR RESPONSE

Thank you for these additional comments, we are very grateful to the reviewers for their attention to detail. We have amended all points as suggested.

P16L52: In reference to previous research done by Whitburn and colleagues, you have mis-typed the findings causing a significant error in the reported results from this study: it should read 'They found that an inability...'

-Amended

I also noted numerous formatting and grammatical issues throughout the manuscript - see file attached.

-Amended

VERSION 3 - REVIEW

REVIEWER	Dr Laura Whitburn La Trobe University, Australia
REVIEW RETURNED	27-Nov-2019

GENERAL COMMENTS	I recommend this paper for publication.
---